# Unsupervised Domain Adaptive Segmentation with Single-content Multi-style Generation and Simplified Pseudo-label Selection

✉Xiao Luan[0000−0003−0010−7361] and Zheng Zhang and Weiqiang Wang and Xiongfeng Huang and Yue Zeng

Chongqing University of Posts and Telecommunications, Chongqing, China
`luanxiao@cqupt.edu.cn`

**Abstract.** For abdominal MRI segmentation, it is difficult to extract the rich information due to the lack of annotated MRI scans. To establish a model of abdominal MRI organ segmentation without MRI annotation, researchers have explored unsupervised cross-modality domain adaptation task for abdominal organ segmentation in MRI scans. And our main idea is to rephrase the unsupervised domain adaptive segmentation problem as an image generation problem and a segmentation problem by a two-stage framework. In the first stage, existing methods usually use generative networks to reduce domain gap and cannot consider the intradomain gap of the target domain. To solve this problem, we propose a single-content multi-style generative network to obtain the multi-style of the target domain rather than the average style. In the second stage, we propose a more simplified pseudo-label selection method to use unlabeled MRI scans. Experiments on the FLARE24 challenge Task3 show that, our method achieved an average score of 63.41% and 68.08% for the lesion DSC and NSD on the validation dataset, respectively. The average running time and area under GPU memory-time curve are 10.36s and 13331MB, respectively. Our method not only focuses on the intra-domain gap but also greatly saves resources in the training phase. Our code will be available at https://github.com/ZZhangZZheng/FLARE24-TASK3.

**Keywords:** Segmentation · Semi-supervised learning · Unsupervised domain adaptation.

## 1 Introduction

Abdominal multi-organ segmentation is a basic step in clinical medicine. In recent years, deep learning methods have shown the capability on segmentation tasks. The abdominal segmentation of CT scan has been greatly developed, but the abdominal segmentation of MRI scan has not been fully explored. One important reason for this is the lack of the data from high-quality annotated abdominal MRI scans. One recent promising approach to solve this problem is unsupervised domain adaptation.

The Fast and Low-resource Semi-supervised Abdominal Organ Segmentation Challenge is a competition focused on abdominal efficiency segmentation [17,18,19]. In addition to focusing on segmentation efficiency and resource consumption, it is also explored in other aspects. In previous competitions, researchers have made great progress in abdominal multi-organ segmentation with CT scans. They achieved high accuracy and fast segmentation efficiency under the premise of consuming less memory resources. And MICCAI FLARE 2024 Task3 (Unsupervised Domain Adaptation for Abdominal Organ Segmentation in MRI Scans) is a new task. It aims to construct abdominal MRI organ segmentation models without MRI annotations. It provides 2300 labeled abdominal CT scans and 1000 unlabeled MRI scans, representing the source and target domains, respectively. Additionally, it offers another 100 and 300 MRI scans for validation and testing purposes, respectively.

In order to learn in the domains without annotation, we can learn in a different but related domain with annotation. The domain we have learned is called the source domain and the domain we are going to learn is called the target domain. And this process is called unsupervised domain adaptation. Due to the domain gap, directly utilizing data from different domains remains highly challenging [25]. At present, with the development of generative adversarial network (GAN) [27], cross-domain style transfer has been widely adopted in unsupervised domain adaptation. The generative adversarial network is used to generate labeled source domain images into target domain data, and we can get the labeled target domain data.

Some researchers use a framework that tightly links generation and segmentation, but the framework is often complex and the inference phase is inefficient. DAR-UNet [25] uses a two-stage framework, which separates the generation process from the segmentation training process. Although its segmentation model is also complex, we can improve its generation and segmentation process respectively.

For generative tasks, StyleGAN [12] performs well. However, it and its variants only use one-to-one mapping of the average style of the target domain. For segmentation tasks, nnU-Net [10] has become a widely used baseline model. However, its inference efficiency is low and it cannot effectively make full use of unlabeled data. The FLARE22 winning algorithm [9] improves nnU-Net to achieve a balance between efficiency and accuracy in the inference phase, but it needs to be trained three times during semi-supervised training.

In order to reduce the complexity of the inference process, we design a two-stage framework for the unsupervised domain adaptation segmentation. At the same time, we observe that the MRI scans of the target domain are very different, they are different modals and come from different machines. Therefore, in the first stage, we propose a single-content multi-style generative network. In the second stage, we use the improved nnU-Net of the FLARE22 winning algorithm in the segmentation phase, but further simplify the process of utilizing unlabeled target domain data.

Our main contributions are summarized as follows:

– We design a two-stage framework for the segmentation without target domain annotation.
– To fit the intra-domain gap, we propose a single-content multi-style generative network, which can better adapt to the data distribution in the target domain. At the same time, it plays the role of data augmentation.
– We use the improved nnU-Net of the FLARE22 winning algorithm for segmentation and simplify its pseudo-label selection method while semi-supervised training.

## 2   Method

As illustrated in Fig. 1, our framework consists of two stages. The first stage uses a single-content multi-style generative network to obtain labeled target domain data. The second stage uses the improved nnU-Net and a simplified pseudo-label selection method for segmentation.

### 2.1   Preprocessing

In the first stage, before feeding the data into the single-content multi-style generative network, we crop the CT scans according to the label to remove the excess background area. At the same time, we normalized the CT scans, limiting their value to between -350 and 350, and scaled them to between 0 and 1. In order to meet the requirement of our anisotropic architecture, the dataset was first spatially normalized to the spacing of [4, 1, 1]. Finally we change them to the same resolution (512,512). For MRI scans, we normalized their intensity and filled or cropped them to the same resolution (512,512). In the second stage, we used the same preprocessing as the improved nnU-Net of the FLARE22 winning algorithm.

### 2.2   Proposed Method

We divide the unsupervised domain adaptation segmentation into two stages. In the first stage, we use our single-content multi-style generative network to reduce domain gap. In the second stage, although its semi-supervised training is complex, the improved nnU-Net of the FLARE22 winning algorithm can already be used as a baseline. So we used the improved nnU-Net with a simpler pseudo-label selection method as our segmentation network.

**The use of labels and images:** We used only CT labeled images and MR unlabeled images. CT unlabeled images were not used. And, we use the FLARE22 winning algorithm but do not use the pseudo labels generated by the FLARE22 winning algorithm [9] and the best-accuracy-algorithm [22].

**The strategies to improve model inference:** A single-stage framework with generation and segmentation tightly linked would reduces the efficiency of inference. Hence, we propose the two-stage framework to improve the inference efficiency. And, the improved nnU-Net of the FLARE22 winning algorithm can

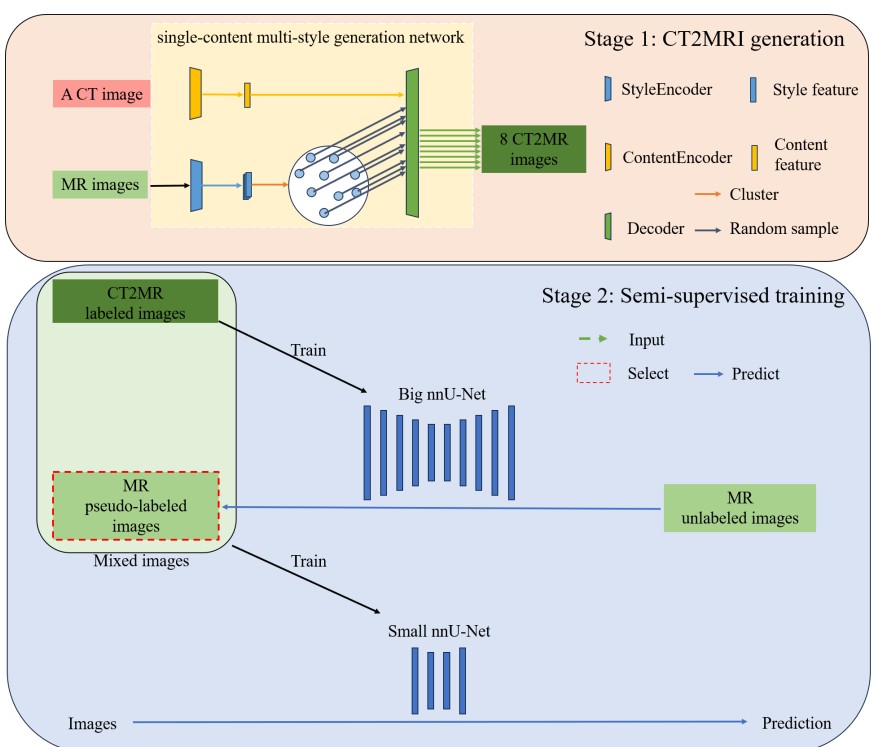

**Fig. 1.** The architecture of the proposed network. Our framework consists of two stages. In the first stage, we train a single-content multi-style generative network that generates 8 CT2MRimages with one CTimage. In the second stage, we use the improved nnU-Net of the FLARE22 winning algorithm as the segmentation network. However, we only train the big nnU-Net once instead of three times by using a simplified pseudo-label selection method.

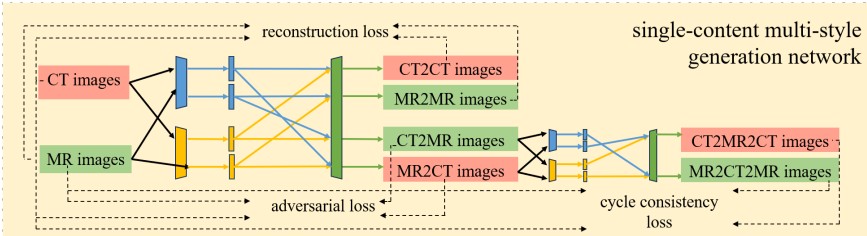

**Fig. 2.** The architecture of the proposed single-content multi-style generative network.

already be used as a baseline, so we optimize the segmentation training efficiency instead of the segmentation inference efficiency.

**Single-content multi-style generative network:** Previous work used a simple style transfer generative network. Researchers usually use a model to generate data corresponding to the average style of the target domain. For FLARE 2024 TASK 3, we can average the MRI target domains into single-style and then generate single-style corresponding CT2MRI data for the CT source domains. However, MRI scans come from different imaging protocols and different machines. Therefore, we propose the multi-style transfer generative network according to the principle of feature decoupling to learn the different styles of the dataset.

As shown in Fig. 2, in the generative network training phase, we train a network consists of a ContentEncoder, a StyleEncoder and a Decoder. CT scans will be passed through ContentEncoder, StyleEncoder to obtain CT content features, CT style features. These features will be passed through the decoder to obtain the reconstructed CT2CT scans. Similarly, we will obtain reconstructed MRI2MRI scans. We will also use CT content features and MRI style features to obtain CT2MRI scans and the same method is used to obtain MRI2CT scans. Similar to above, but we are not reconstructing the scans. We only extract the content and style features of the CT2MRI,MRI2CT scans and generate CT2MRI2CT,MRI2CT2MRI scans. Similarly for MRI scans, but using CT scans as an example, we will optimize the ContentEncoder, the StyleEncoder and the Decoder by calculating the reconstruction loss between CT scans and CT2CT scans, the adversarial loss between CT scans and MRI2CT, and the cycle consistency loss between CT scans and CT2MRI2CT. It is important to state that our network operates on a single layer of the scans, which also occurs in the inference phase. And the reconstruction loss, the adversarial loss and the cycle consistency loss are shown follows:

$$\mathcal{L}_{rec} = E_{I_a \sim \mathcal{I}_a} \left\| I_a - \mathcal{G}_{dec}(\mathcal{G}_{enc_c}(I_a), \mathcal{G}_{enc_{sa}}(I_a)) \right\|_1 \tag{1}$$

$$\mathcal{L}_{GAN} = E_{c_b \sim \mathcal{C}_b, s_a \sim \mathcal{S}_a}[\log(1 - \mathcal{D}_a(\mathcal{G}_{dec}(c_b, s_a)))] + E_{I_a \sim \mathcal{I}_a}[\log(\mathcal{D}_a(I_a))]. \tag{2}$$

$$\mathcal{L}_{cycle} = E_{I_a \sim \mathcal{I}_a} \left\| I_a - I_{aba} \right\|_1 \tag{3}$$

$I_a$ refers to the CT or MR images, $I_{aba}$ refers to the CT2MR2CT or MR2CT2MR images. $\mathcal{G}_{enc_c}()$ refers to the ContentEncode, $\mathcal{G}_{enc_{sa}}()$ refers to the StyleEncode and $\mathcal{G}_{dec}()$ refers to the Decoder.

We jointly train the encoders, decoders, and discriminator to optimize the final objective

$$\min_{\mathcal{G}} \max_{\mathcal{D}} \mathcal{L}(\mathcal{G}_{enc_c}, \mathcal{G}_{enc_{sa}}, \mathcal{G}_{enc_{sb}}, \mathcal{G}_{dec}, \mathcal{D}_a) = \mathcal{L}_{rec} + \mathcal{L}_{GAN} + \mathcal{L}_{cycle}. \tag{4}$$

In the generative network inference phase. We first use the StyleEncoder to obtain style features for all MRI scans and then cluster these style features. The number of clusters here is 8. For each CT content feature obtained from the ContentEncoder, and one random of each type of style feature is synthesized to

obtain the generated CT2MRI scan. Our single-content multi-style generative network enables a single CT scan to become 8 CT2MRI scans. These CT2MRI scans will be used for the next stage.

**Segmentation network:** The nnU-Net is a useful baseline for medical segmentation tasks. However, it is complex and time-consuming. And the improved nnU-Net of the FLARE22 winning algorithm make it simple and efficient, so we use the improved nnU-Net as our segmentation network. Although the improved nnU-Net of the FLARE22 winning algorithm is very fast in the inference phase, it needs to be trained three times to perform uncertain calculations, which undoubtedly consumes resources. We hope to further conserve resources.

**Simplified pseudo-label selection method:** Our motivation is simple. Through careful observation of the pseudo-labels, we identified that the majority of errors manifest as undersegmentation. Notably, these undersegmented results consistently exhibit smaller file sizes compared to accurate segmentations. So we have a simpler pseudo-labels selection strategy. We discarded 1348 MRI scans with pseudo-labels smaller than 14KB as unreliable. Our method trains just once instead of three times. This pseudo-labels selection method is not as significantly improved as the semi-supervised method of the FLARE22 winning algorithm, but it is simpler.

**Loss function:** We use the summation of the reconstruction loss, cycle consistency loss and adversarial loss because these have been shown to help decouple content and style as well as final generation [25].

We use the summation between Dice loss and cross-entropy loss because compound loss functions have been proven to be robust in various medical image segmentation tasks [14].

### 2.3   Post-processing

In order to conserve resources, we do not use any post-processing in our training and inference.

## 3   Experiments

### 3.1   Dataset and evaluation measures

The training dataset is curated from more than 30 medical centers under the license permission, including TCIA [3], LiTS [2], MSD [21], KiTS [7,8], autoPET [6,5], AMOS [11], LLD-MMRI [13], TotalSegmentator [23], and Abdomen-CT-1K [20], and past FLARE Challenges [17,18,19]. The training set includes 2050 abdomen CT scans and over 4000 MRI scans. The validation and testing sets include 110 and 300 MRI scans, respectively, which cover various MRI sequences, such as T1, T2, DWI, and so on. The organ annotation process used ITK-SNAP [26], nnU-Net [10], MedSAM [15], and Slicer Plugins [4,16].

The evaluation metrics encompass two accuracy measures—Dice Similarity Coefficient (DSC) and Normalized Surface Dice (NSD)—alongside two efficiency

measures—running time and area under the GPU memory-time curve. These metrics collectively contribute to the ranking computation. Furthermore, the running time and GPU memory consumption are considered within tolerances of 15 seconds and 4 GB, respectively.

### 3.2 Implementation details

**Environment settings** The development environments and requirements are presented in Table 1.

**Table 1.** Development environments and requirements.

| | |
|---|---|
| System | Ubuntu 22.04.4 LTS |
| CPU | Intel(R) Core(TM) i9-13900K CPU@5.80GHz |
| RAM | 4×32GB; 2133MT/s |
| GPU (number and type) | Two NVIDIA RTX 3090 24G |
| CUDA version | 12.2 |
| Programming language | Python 3.7.12 |
| Deep learning framework | torch 1.13.1, torchvision 0.14.1 |
| Specific dependencies | nnU-Net 1.7.0 |
| Code | https://github.com/ZZhangZZheng/FLARE24-TASK3 |

**Training protocols** The training protocols of single-content multi-style generative network, big nnU-Net and small nnU-Net are listed in Table 2, Table 3 and Table 4 respectively.

In the first stage, we input each slice (1×512×512) of the preprocessed 3D data into the generative network without any data augmentation [17] for the training. Our single-content multi-style generative network not only closes the distance between the source and target domains but also reduces the intra-domain gap. And we obtain 400 CT2MRI labeled scans through 50 CT labeled scans for the second stage of segmentation.

For the segmentation task in the second stage, we used the same data augmentation strategy as the FLARE22 winning algorithm for the 400 CT2MRI labeled scans. Such as additive brightness, gamma, rotation, scaling, and elastic deformation. And the patch sampling strategy is also consistent with it. But we simplified the optimal model selection criteria. After completing the training of CT2MRI only once images with big nnU-Net, we directly chose it as the optimal model to reduce resource usage. Then we used the big nnU-Net get the pseudo-labels for the MRI unlabeled scans. And we removed the 1348 low-quality scans through the simplified pseudo-label selection method, and continued training to obtain the small nnU-Net.

We calculated the number of model parameters, number of flops and the carbon footprint [1] of the single-content multi-style generative network, the big nnU-Net and the small nnU-Net through the tools.

**Table 2.** Training protocols for single-content multi-style generative network.

| Network initialization | |
|---|---|
| Batch size | 1 |
| Patch size | 1×512×512 |
| Total epochs | 100 |
| Optimizer | AdaBelief |
| Discriminators initial learning rate (lr) | 0.0002 |
| Generators initial learning rate (lr) | 0.0001 |
| Training time | 12 hours |
| Loss function | Reconstruction loss, cycle consistency loss and adversarial loss |
| Number of model parameters | 84 M[1] |
| Number of flops | 100 G[2] |
| $CO_2$eq | 4.15 Kg[3] |

[1] https://github.com/sksq96/pytorch-summary
[2] https://github.com/facebookresearch/fvcore
[3] https://github.com/lfwa/carbontracker/

**Table 3.** Training protocols for big nnU-Net.

| Network initialization | |
|---|---|
| Batch size | 2 |
| Patch size | 48×224×224 |
| Total epochs | 1000 |
| Optimizer | SGD with nesterov momentum (μ = 0.99) |
| Initial learning rate (lr) | 0.01 |
| Lr decay schedule | Poly learning rate policy: $(1 - epoch/1000)^{0.9}$ |
| Training time | 36 hours |
| Loss function | Dice loss and cross entropy loss |
| Number of model parameters | 82 M |
| Number of flops | 776 G |
| $CO_2$eq | 10.48 Kg |

**Table 4.** Training protocols for small nnU-Net.

| Network initialization | |
|---|---|
| Batch size | 2 |
| Patch size | 32×128×192 |
| Total epochs | 1500 |
| Optimizer | SGD with nesterov momentum ($\mu = 0.99$) |
| Initial learning rate (lr) | 0.01 |
| Lr decay schedule | Poly learning rate policy: $(1 - epoch/1500)^{0.9}$ |
| Training time | 16 hours |
| Loss function | Dice loss and cross entropy loss |
| Number of model parameters | 5.4 M |
| Number of flops | 136 G |
| $CO_2$eq | 1.87 Kg |

## 4 Results and discussion

We use the trained small nnU-Net to predict the segmentation. We list the results of DSC and NSD on the validation dataset in Table 5.

**Table 5.** Quantitative evaluation results. The validation denotes the performance on the 110 validation cases with ground truth.

| Target | Validation | |
|---|---|---|
| | DSC(%) | NSD(%) |
| Liver | $84.65 \pm 18.51$ | $81.95 \pm 19.91$ |
| Right kidney | $78.25 \pm 29.04$ | $77.36 \pm 30.17$ |
| Spleen | $74.07 \pm 32.40$ | $74.29 \pm 33.92$ |
| Pancreas | $54.17 \pm 31.17$ | $63.16 \pm 37.54$ |
| Aorta | $75.92 \pm 27.08$ | $78.52 \pm 29.02$ |
| Inferior vena cava | $53.96 \pm 32.13$ | $54.12 \pm 33.92$ |
| Right adrenal gland | $47.85 \pm 25.21$ | $61.76 \pm 37.54$ |
| Left adrenal gland | $52.13 \pm 28.67$ | $64.51 \pm 34.87$ |
| Gallbladder | $65.69 \pm 34.31$ | $63.13 \pm 34.83$ |
| Esophagus | $46.62 \pm 27.05$ | $56.61 \pm 33.69$ |
| Stomach | $64.20 \pm 28.69$ | $66.18 \pm 31.90$ |
| Duodenum | $42.21 \pm 27.03$ | $57.54 \pm 37.25$ |
| Left kidney | $84.58 \pm 21.45$ | $85.97 \pm 23.00$ |
| Average | $63.41 \pm 19.95$ | $68.08 \pm 22.93$ |

### 4.1 Quantitative results on validation set

In order to utilize unlabeled data, we used the method of generating pseudo-labels. However, unlike the FLARE22 winning algorithm training 3 times, which

uses uncertainty to filter out pseudo-labels. We filter out unreliable 1348 MRI scans based on the size of the pseudo-label. That is, we have 400 CT2MRI labeled images and 3469 MR images with reliable pseudo labels in the end.

**Ablation study** We list the results of DSC and NSD on the validation dataset in Table 6. As shown in Table 6, the direct application of semi-supervised learning led to model performance degradation, particularly for organs with initially poor segmentation results. We argue this stems primarily from segmentation errors (mostly undersegmentation). Our proposed selection strategy demonstrates robust performance, confirming that the pseudo-label filtering method based on file size effectively removes undersegmented pseudo-labels. By using this method, we improved our average DSC by 2.8% and NSD by 3.63%.

**Table 6.** Comparison of Without-Semi, Semi-supervised, and Semi-with-selection Methods, where Without-Semi enotes training using only CT2MRI data. Semi-supervised denotes training using pseudo-labeled unlabeled data and CT2MRI data. Semi-with-selection denotes training using selected pseudo-labeled unlabeled data and CT2MRI data.

| Target | Without-Semi | | Semi-supervised | | Semi-with-selection | |
|---|---|---|---|---|---|---|
| | DSC(%) | NSD(%) | DSC(%) | NSD(%) | DSC(%) | NSD (%) |
| Liver | 83.75 | 80.12 | 91.06 | 87.58 | 84.65 | 81.95 |
| Right kidney | 76.05 | 74.47 | 78.50 | 74.77 | 78.25 | 77.36 |
| Spleen | 71.68 | 72.66 | 60.00 | 59.59 | 74.07 | 74.29 |
| Pancreas | 52.24 | 59.59 | 50.63 | 59.85 | 54.17 | 63.16 |
| Aorta | 76.23 | 78.67 | 74.41 | 77.89 | 75.92 | 78.52 |
| Inferior vena cava | 50.64 | 49.64 | 36.49 | 33.70 | 53.96 | 54.12 |
| Right adrenal gland | 46.58 | 60.44 | 30.70 | 40.35 | 47.85 | 61.76 |
| Left adrenal gland | 46.59 | 56.65 | 26.56 | 34.84 | 52.13 | 64.51 |
| Gallbladder | 58.99 | 56.60 | 55.43 | 52.41 | 65.69 | 63.13 |
| Esophagus | 43.86 | 52.09 | 31.83 | 39.20 | 46.62 | 56.61 |
| Stomach | 59.92 | 62.12 | 52.05 | 53.64 | 64.20 | 66.18 |
| Duodenum | 41.80 | 55.35 | 42.44 | 59.29 | 42.21 | 57.54 |
| Left kidney | 79.63 | 79.48 | 78.26 | 77.42 | 84.58 | 85.97 |
| Average | 60.61 | 64.45 | 54.49 | 57.73 | 63.41 | 68.08 |

## 4.2  Qualitative results on validation set

Fig. 3 shows 4 representative segmentation results of our small nnU-Net trained on 50 labeled data and 3469 selected pseudo labels for final submission. For Case #amos_0600 and Case #amos_7562, it is easy to see that some under-segmentation and over-segmentation errors occurred. For Case #amos_7891 and Case #amos_0581, the network has better segmentation performance, but there is still under-segmentation errors. We argue that this is due to poor performance of big nnU-Net because the generation is not good enough, even though

a large amount of unlabeled data improves the under-segmentation and over-segmentation errors.

**Table 7.** Quantitative evaluation of segmentation efficiency in terms of the running them and GPU memory consumption. Total GPU denotes the area under GPU Memory-Time curve. Evaluation GPU platform: NVIDIA QUADRO RTX5000 (16G).

| Case ID | Image Size | Running Time (s) | Max GPU (MB) | Total GPU (MB) |
|---|---|---|---|---|
| amos_0540 | (192, 192, 100) | 9.67 | 2587 | 11951 |
| amos_7324 | (256, 256, 80) | 9.71 | 3533 | 11945 |
| amos_0507 | (320, 290, 72) | 13.38 | 3503 | 12464 |
| amos_7236 | (400, 400, 115) | 10.28 | 3079 | 13434 |
| amos_7799 | (432, 432, 40) | 10.27 | 2719 | 13686 |
| amos_0557 | (512, 152, 512) | 11.38 | 4697 | 15372 |
| amos_0546 | (576, 468, 72) | 10.12 | 2667 | 12932 |
| amos_8082 | (1024, 1024, 82) | 12.7 | 2713 | 17716 |

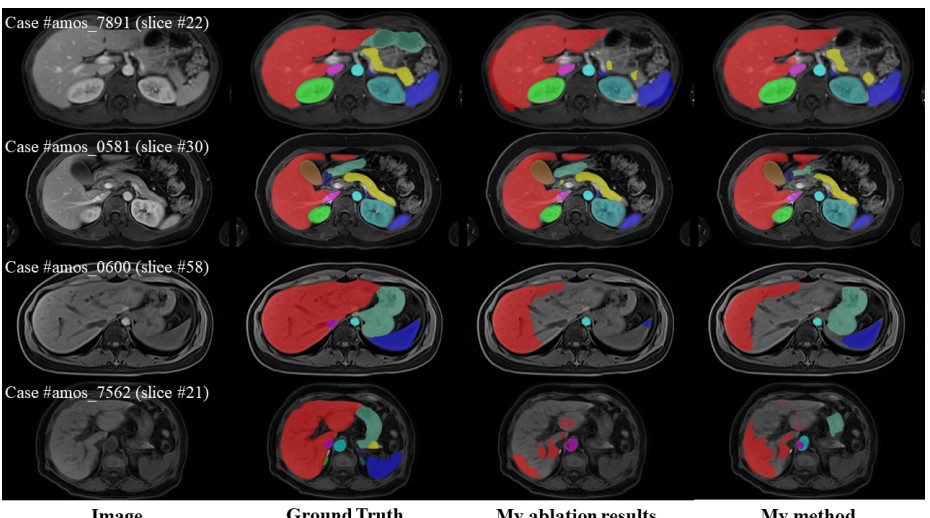

**Fig. 3.** We show two examples with good segmentation results and two examples with bad segmentation results in the validation set. Our method denotes the final small nnU-Net by training pseudo-labeled MRI scans and generated CT2MRI scans. Our ablation results denote the small nnU-Net by only training generated CT2MRI scans.

### 4.3   Segmentation efficiency results on validation set

We build our small nnU-Net as a docker image for final submission. In Table 7, we list our segmentation efficiency results on validation set during the challenge. The results feedback from the challenge organizer.

### 4.4   Results on final testing set

The final testing results for the proposed method in the FLARE 2024 challenge are summarized in Table 8. The table presents the performance metrics of the method, including the Dice Similarity Coefficient (DSC), Normalized Surface Distance (NSD), inference time, and GPU memory usage. Each metric is re ported with both the mean and standard deviation (Mean ± Std), as well as the median along with the first and third quartiles (Median (Q1, Q3)).

**Table 8.** Final testing results of the proposed method on the FLARE 2024 challenge.

| Metric | Mean ± Std | Median (Q1, Q3) |
| --- | --- | --- |
| DSC (%) | 43.1 ± 33.8 | 53.6 (2.3,76.3) |
| NSD (%) | 45.4 ± 36.1 | 56.6 (2.2,80.4) |
| Inference Time (s) | 10.4 ± 1.4 | 9.9 (9.8,10.2) |
| GPU Memory (MB) | 573657.3 ± 84163.7 | 548777.6 (534456.4,568452.1) |

### 4.5   Limitation and future work

Although our single-content multi-style generative network can better learn multi-center data, the quality of generation is still a limiting factor. Pseudo-labels are widely used in semi-supervision tasks, in which denoising is the key. We will refer to the updated research progress to improve the quality of generated images and pseudo labels in our future work.

## 5   Conclusion

In this paper, we propose a two-stage framework to unsupervised domain adaptation. Our two-stage framework makes inference simple by turning the unsupervised domain-adapted segmentation problem into two problems: cross-domain generation and semi-supervised segmentation. In the first stage, we used a single-content multi-style generative network to learn multi-center data which focuses on the intra-domain gap of the target domain while achieving cross-domain generation. In the second stage, we use the improved nnU-Net of the FLARE22 winning algorithm and further simplified the semi-supervised method. And our simplified pseudo-label selection method makes training more efficient. We hope our research can help others.

**Acknowledgements** The authors of this paper declare that the segmentation method they implemented for participation in the FLARE 2024 challenge has not used any pre-trained models nor additional datasets other than those provided by the organizers. The proposed solution is fully automatic without any manual intervention. We thank all data owners for making the CT scans publicly available and CodaLab [24] for hosting the challenge platform.

## Disclosure of Interests

The authors declare no competing interests.

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

**Table 9.** Checklist Table. Please fill out this checklist table in the answer column.

| Requirements | Answer |
| --- | --- |
| A meaningful title | Yes |
| The number of authors ($\leq$6) | 5 |
| Author affiliations and ORCID | Yes |
| Corresponding author email is presented | Yes |
| Validation scores are presented in the abstract | Yes |
| Introduction includes at least three parts: background, related work, and motivation | Yes |
| A pipeline/network figure is provided | Fig 1 |
| Pre-processing | 3 |
| Strategies to use the partial label | 3 |
| Strategies to use the unlabeled images. | 6 |
| Strategies to improve model inference | 3 |
| Post-processing | 6 |
| The dataset and evaluation metric section are presented | 6 |
| Environment setting table is provided | Table 1 |
| Training protocol table is provided | Table 2, Table 3 and Table 4 |
| Ablation study | 10 |
| Efficiency evaluation results are provided | Table 7 |
| Visualized segmentation example is provided | Figure 3 |
| Limitation and future work are presented | Yes |
| Reference format is consistent. | Yes |

Response to Reviewer n39Y

We sincerely thank Reviewer n39Y for the valuable comments. Below are our point-by-point responses addressing the concerns raised.

Comment 1: Fig. 1 is too complex to understand. The authors could make it concise for high level ideas or split it to serveral figures.

Thank you for pointing out the limitations of Fig. 1. In response to the reviewer's comments, we have reorganized the original figure into two distinct figures. The revised Fig. 1 provides a clear overview of the high level ideas, while Fig. 2 specifically details the generative network architecture.

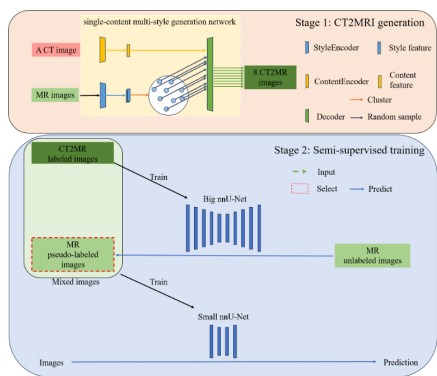

**Fig. 1.** The architecture of the proposed network. Our framework consists of two stages. In the first stage, we train a single-content multi-style generative network that generates 8 CT2MRimages with one CTimage. In the second stage, we use the improved nnU-Net of the FLARE22 winning algorithm as the segmentation network. However, we only train the big nnU-Net once instead of three times by using a simplified pseudo-label selection method.

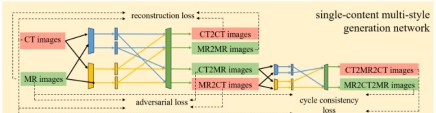

**Fig. 2.** The architecture of the proposed single-content multi-style generative network.

Comment 2: The loss functions should be explained in details.

Thank you for pointing out the need for the details of the loss functions. We have provided additional clarification in Section 2.2.

$$\mathcal{L}_{rec} = E_{I_a \sim \mathcal{I}_a} \|I_a - \mathcal{G}_{dec}(\mathcal{G}_{enc_c}(I_a), \mathcal{G}_{enc_{sa}}(I_a))\|_1 \quad (1)$$

$$\mathcal{L}_{GAN} = E_{c_b \sim \mathcal{C}_b, s_a \sim \mathcal{S}_a}[\log(1 - \mathcal{D}_a(\mathcal{G}_{dec}(c_b, s_a)))] + E_{I_a \sim \mathcal{I}_a}[\log(\mathcal{D}_a(I_a))]. \quad (2)$$

$$\mathcal{L}_{cycle} = E_{I_a \sim \mathcal{I}_a} \|I_a - I_{aba}\|_1 \quad (3)$$

$I_a$ refers to the CT or MR images, $I_{aba}$ refers to the CT2MR2CT or MR2CT2MR images. $\mathcal{G}_{enc_c}()$ refers to the ContentEncode, $\mathcal{G}_{enc_{sa}}()$ refers to the StyleEncode and $\mathcal{G}_{dec}()$ refers to the Decoder.

We jointly train the encoders, decoders, and discriminators to optimize the final objective, which is a weighted sum of the adversarial loss and the reconstruction loss terms

$$\min_{\mathcal{G}} \max_{\mathcal{D}} \mathcal{L}(\mathcal{G}_{enc_c}, \mathcal{G}_{enc_{sa}}, \mathcal{G}_{enc_{sb}}, \mathcal{G}_{dec}, \mathcal{D}_a) = \mathcal{L}_{rec} + \mathcal{L}_{GAN} + \mathcal{L}_{cycle}. \quad (4)$$

Comment 3: There are many typos or formatting issues. In the caption of Figure 2, "My ablation results denotes" should be "My ablation results denote".

Thank you very much for your careful review and for pointing out the typos or formatting issues. Specifically, we have revised the sentence as follows:"Our ablation results denote the small nnU-Net by only training generated CT2MRI scans."

Comment 4: "Our method need train just once instead of three times." in the "Simplified pseudo-label selection method" should be "Our method needs to train just once instead of three times.".

Thank you very much for your careful review and for pointing out the typos or formatting issues. We sincerely apologize for this oversight. Upon reviewing the manuscript, we have made the necessary correction as per your suggestion. Specifically, we have revised the sentence as follows:"Our method needs train just once instead of three times."

We are deeply grateful for your thorough evaluation. Your expert comments have been instrumental in enhancing the scholarly quality of this work. We remain committed to refining all details to achieve the highest standards of academic rigor.

Response to Reviewer RzQq

We sincerely thank Reviewer RzQq for the valuable comments. Below are our point-by-point responses addressing the concerns raised.

Comment 1: As for the method description of single-content multi-style generation network, it is not clear and cannot correspond well with Figure 1. It is suggested to modify the presentation logic and layout of the graph, and add necessary formulas.

Thank you for pointing out this issue. We have shown the single-content multi-style generation network in Fig. 2 and add the additional formulas.

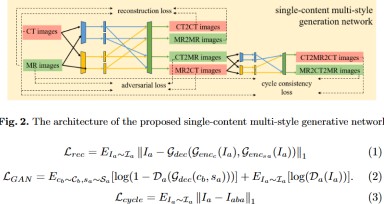

**Fig. 2.** The architecture of the proposed single-content multi-style generative network.

$$\mathcal{L}_{rec} = E_{I_a \sim \mathcal{I}_a} \left\| I_a - \mathcal{G}_{dec}(\mathcal{G}_{enc_c}(I_a), \mathcal{G}_{enc_{s_a}}(I_a)) \right\|_1 \qquad (1)$$

$$\mathcal{L}_{GAN} = E_{c_b \sim \mathcal{C}_b, s_a \sim \mathcal{S}_a}[\log(1 - \mathcal{D}_a(\mathcal{G}_{dec}(c_b, s_a)))] + E_{I_a \sim \mathcal{I}_a}[\log(\mathcal{D}_a(I_a))]. \qquad (2)$$

$$\mathcal{L}_{cycle} = E_{I_a \sim \mathcal{I}_a} \left\| I_a - I_{aba} \right\|_1 \qquad (3)$$

$I_a$ refers to the CT or MR images, $I_{aba}$ refers to the CT2MR2CT or MR2CT2MR images. $\mathcal{G}_{enc_c}()$ refers to the ContentEncode, $\mathcal{G}_{enc_{s_a}}()$ refers to the StyleEncode and $\mathcal{G}_{dec}()$ refers to the Decoder.

We jointly train the encoders, decoders, and discriminators to optimize the final objective, which is a weighted sum of the adversarial loss and the reconstruction loss terms

$$\min_{\mathcal{G}} \max_{\mathcal{D}} \mathcal{L}(\mathcal{G}_{enc_c}, \mathcal{G}_{enc_{s_a}}, \mathcal{G}_{enc_{s_b}}, \mathcal{G}_{dec}, \mathcal{D}_a) = \mathcal{L}_{rec} + \mathcal{L}_{GAN} + \mathcal{L}_{cycle}. \qquad (4)$$

Comment 2: There is a lack of ablation experiments and insufficient analysis of experiments.

Thank you for your thoughtful and constructive comment. We have added the relevant ablation experiments in Table 6 and provided analysis of experiments.

"As shown in Table 6, the direct application of semi-supervised learn- ing led to model performance degradation, particularly for organs with initially poor segmentation results. We argue this stems primarily from segmentation er- rors (mostly undersegmentation). Our proposed selection strategy demonstrates ro- bust performance, confirming that the pseudo-label filtering method based on file size effectively removes undersegmented pseudo-labels."

**Table 6.** Comparison of Without-Semi, Semi-supervised, and Semi-with-selection Methods, where Without-Semi enotes training using only CT2MRI data. Semi-supervised denotes training using pseudo-labeled unlabeled data and CT2MRI data. Semi-with-selection denotes training using selected pseudo-labeled unlabeled data and CT2MRI data.

| Target | Without-Semi | | Semi-supervised | | Semi-with-selection | |
|---|---|---|---|---|---|---|
| | DSC(%) | NSD(%) | DSC(%) | NSD(%) | DSC(%) | NSD (%) |
| Liver | 83.75 | 80.12 | 91.06 | 87.58 | 84.65 | 81.95 |
| Right kidney | 76.05 | 74.47 | 78.50 | 74.77 | 78.25 | 77.36 |
| Spleen | 71.68 | 72.66 | 60.00 | 59.59 | 74.07 | 74.29 |
| Pancreas | 52.24 | 59.59 | 50.63 | 59.85 | 54.17 | 63.16 |
| Aorta | 76.23 | 78.67 | 74.41 | 77.89 | 75.92 | 78.52 |
| Inferior vena cava | 50.64 | 49.64 | 36.49 | 33.70 | 53.96 | 54.12 |
| Right adrenal gland | 46.58 | 60.44 | 30.70 | 40.35 | 47.85 | 61.76 |
| Left adrenal gland | 46.59 | 56.65 | 26.56 | 34.84 | 52.13 | 64.51 |
| Gallbladder | 58.99 | 56.60 | 55.43 | 52.41 | 65.69 | 63.13 |
| Esophagus | 43.86 | 52.09 | 31.83 | 39.20 | 46.62 | 56.61 |
| Stomach | 59.92 | 62.12 | 52.05 | 53.64 | 64.20 | 66.18 |
| Duodemum | 41.80 | 55.35 | 42.44 | 59.29 | 42.21 | 57.54 |
| Left kidney | 79.63 | 79.48 | 78.26 | 77.42 | 84.58 | 85.97 |
| Average | 60.61 | 64.45 | 54.49 | 57.73 | 63.41 | 68.08 |

Response to Reviewer RLV9

We sincerely thank Reviewer RLV9 for the valuable comments. Below are our point-by-point responses addressing the concerns raised.

Comment 1: Figure 1 is somewhat complex, and the distinction between modules within the two stages is unclear. It is recommended to organize the colors and layout of the modules.

Thank you for pointing out this issue. We appreciate your suggestions for improving the Fig. 1. As per your recommendation, we have reorganized the colors and layout of the modules. Specifically, we separated the single-content multi-style generative network from Fig. 1 and assigned distinct colors to different modules and multimodal data.

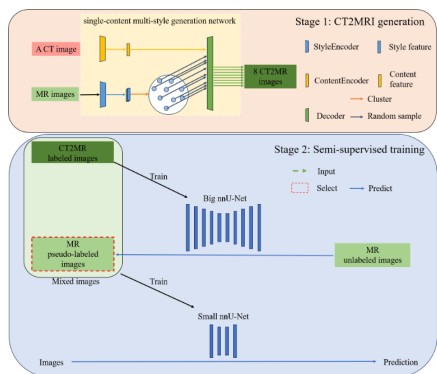

**Fig. 1.** The architecture of the proposed network. Our framework consists of two stages. In the first stage, we train a single-content multi-style generative network that generates 8 CT2MRimages with one CTimage. In the second stage, we use the improved nnU-Net of the FLARE22 winning algorithm as the segmentation network. However, we only train the big nnU-Net once instead of three times by using a simplified pseudo-label selection method.

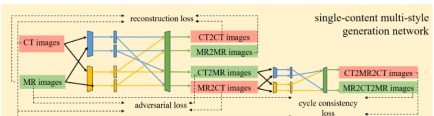

**Fig. 2.** The architecture of the proposed single-content multi-style generative network.

We sincerely apologize for the omission of ablation experiments to validate the effectiveness of the style transfer network. The effectiveness of the style transfer network has been validated by others and we acknowledge the insufficient introduction of background knowledge in the original manuscript. We have now supplemented the Introduction section as follows: "Due to the domain gap, directly utilizing data from different domains remains highly challenging ~\cite{yao2022novel}. At present, with the development of generative adversarial network (GAN)~\cite{zhu2017unpaired}, cross-domain style transfer has been widely adopted in unsupervised domain adaptation."

Comment 3: The pseudo-label filtering method based on file size is not well explained, and there is a lack of ablation experiments comparing using pseudo-labels directly versus using the proposed filtering method to validate the effectiveness of the pseudo-label filtering approach presented in this paper.

Thank you for your thoughtful and constructive comment. your concern about the pseudo-label filtering method based on file size, we have carefully address this issue. In Section 2.2, we provided a better explanation and added the relevant ablation experiments in Table 6.

"Our motivation is simple. Through careful observation of the pseudo-labels, we identified that the majority of errors manifest as undersegmentation. In particular, these under-segmented results consistently exhibit smaller file sizes compared to accurate segmentations."

**Table 6.** Comparison of Without-Semi, Semi-supervised, and Semi-with-selection Methods, where Without-Semi enotes training using only CT2MRI data. Semi-supervised denotes training using pseudo-labeled unlabeled data and CT2MRI data. Semi-with-selection denotes training using selected pseudo-labeled unlabeled data and CT2MRI data.

| Target | Without-Semi | | Semi-supervised | | Semi-with-selection | |
|---|---|---|---|---|---|---|
| | DSC(%) | NSD(%) | DSC(%) | NSD(%) | DSC(%) | NSD (%) |
| Liver | 83.75 | 80.12 | 91.06 | 87.58 | 84.65 | 81.95 |
| Right kidney | 76.05 | 74.47 | 78.50 | 74.77 | 78.25 | 77.36 |
| Spleen | 71.68 | 72.66 | 60.00 | 59.59 | 74.07 | 74.29 |
| Pancreas | 52.24 | 59.59 | 50.63 | 59.85 | 54.17 | 63.16 |
| Aorta | 76.23 | 78.67 | 74.41 | 77.89 | 75.92 | 78.52 |
| Inferior vena cava | 50.64 | 49.64 | 36.49 | 33.70 | 53.96 | 54.12 |
| Right adrenal gland | 46.58 | 60.44 | 30.70 | 40.35 | 47.85 | 61.76 |
| Left adrenal gland | 46.59 | 56.65 | 26.56 | 34.84 | 52.13 | 64.51 |
| Gallbladder | 58.99 | 56.60 | 55.43 | 52.41 | 65.69 | 63.13 |
| Esophagus | 43.86 | 52.09 | 31.83 | 39.20 | 46.62 | 56.61 |
| Stomach | 59.92 | 62.12 | 52.05 | 53.64 | 64.20 | 66.18 |
| Duodenum | 41.80 | 55.35 | 42.44 | 59.29 | 42.21 | 57.54 |
| Left kidney | 79.63 | 79.48 | 78.26 | 77.42 | 84.58 | 85.97 |
| Average | 60.61 | 64.45 | 54.49 | 57.73 | 63.41 | 68.08 |