# OpenReview forum: "Unsupervised Domain Adaptive Segmentation with Single-content Multi-style Generation and Simplified Pseudo-label Selection"
_MICCAI.org/2024/Challenge/FLARE — FLARE 2024 withMinorRevisions_

### Official Review · Reviewer_RLV9 · 2025-01-17
**Review of "Unsupervised Domain Adaptive Segmentation with Single-content Multi-style Generation and Simplified Pseudo-label Selection"**

**Rating:** 7
**Confidence:** 4

**Review:**

The paper proposes a one-to-many domain style transfer network combined with semi-supervised multi-organ segmentation, and reduces resource consumption by using a small nnU-Net. The writing is clear and easy to understand.

### Strengths:
1. The paper proposes a simple and direct one-to-many domain style transfer network combined with semi-supervised multi-organ segmentation.
2. A simple and fast pseudo-label filtering method based on file size is proposed.
3. The provided ablation experiments confirm that pseudo-labels improve the model's performance.

### Areas for Improvement:
1. Figure 1 is somewhat complex, and the distinction between modules within the two stages is unclear. It is recommended to organize the colors and layout of the modules.
2. There is a lack of ablation experiments using labeled CT samples and unlabeled MR samples for semi-supervised training to confirm the effectiveness of the style transfer network.
3. The pseudo-label filtering method based on file size is not well explained, and there is a lack of ablation experiments comparing using pseudo-labels directly versus using the proposed filtering method to validate the effectiveness of the pseudo-label filtering approach presented in this paper.

---

> ### Author Response · Authors · 2025-03-29
> **Response to Reviewer RLV9**
>
> Comment 1: Figure 1 is somewhat complex, and the distinction between modules within the two stages is unclear. It is recommended to organize the colors and layout of the modules.
>
> Thank you for pointing out this issue. We appreciate your suggestions for improving the Fig. 1. As per your recommendation, we have reorganized the colors and layout of the modules. Specifically, we separated the single-content multi-style generative network from Fig. 1 and assigned distinct colors to different modules and multimodal data.
>
> Comment 2: There is a lack of ablation experiments using labeled CT samples and unlabeled MR samples for semi-supervised training to confirm the effectiveness of the style transfer network.
>
> We sincerely apologize for the omission of ablation experiments to validate the effectiveness of the style transfer network. The effectiveness of the style transfer network has been validated by others and we acknowledge the insufficient introduction of background knowledge in the original manuscript. We have now supplemented the Introduction section as follows:
>
> "Due to the domain gap, directly utilizing data from different domains remains highly challenging cite{yao2022novel}. At present, with the development of generative adversarial network (GAN)  cite{zhu2017unpaired}, cross-domain style transfer has been widely adopted in unsupervised domain adaptation."
>
> Comment 3: The pseudo-label filtering method based on file size is not well explained, and there is a lack of ablation experiments comparing using pseudo-labels directly versus using the proposed filtering method to validate the effectiveness of the pseudo-label filtering approach presented in this paper.
>
> Thank you for your thoughtful and constructive comment. your concern about the pseudo-label filtering method based on file size, we have carefully address this issue. In Section 2.2, we provided a better explanation and added the relevant ablation experiments in Table 6.
>
> “Our motivation is simple. Through careful observation of the pseudo-labels, we identified that the majority of errors manifest as undersegmentation. In particular, these under-segmented results consistently exhibit smaller file sizes compared to accurate segmentations."

---

### Official Review · Reviewer_RzQq · 2025-01-25

**Rating:** 7
**Confidence:** 5

**Review:**

This paper proposes a single-content multi-style generative network to fit the intra-domain gap and reduces resource consumption by using a small nnU-Net. The writing is clear and easy to understand. Here are my concerns.

1. As for the method description of single-content multi-style generation network, it is not clear and cannot correspond well with Figure 1. It is suggested to modify the presentation logic and layout of the graph, and add necessary formulas.
2. There is a lack of ablation experiments and insufficient analysis of experiments.

---

> ### Author Response · Authors · 2025-03-29
> **Response to Reviewer RzQq**
>
> Comment 1: As for the method description of single-content multi-style generation network, it is not clear and cannot correspond well with Figure 1. It is suggested to modify the presentation logic and layout of the graph, and add
> necessary formulas.
>
> Thank you for pointing out this issue. We have shown the single-content multi-style generation network in Fig. 2 and add the additional formulas.
>
> Comment 2: There is a lack of ablation experiments and insufficient analysis of experiments.
>
> Thank you for your thoughtful and constructive comment. We have added the relevant ablation experiments in Table 6 and provided analysis of experiments.
> "As shown in Table 6, the direct application of semi-supervised learn- ing led to model performance degradation, particularly for organs with initially poor segmentation results. We argue this stems primarily from segmentation errors (mostly undersegmentation). Our proposed selection strategy demonstrates robust performance, confirming that the pseudo-label filtering method based on file size effectively removes undersegmented pseudo-labels."

---

### Official Review · Reviewer_n39Y · 2025-03-02
**Review of "Unsupervised Domain Adaptive Segmentation with Single-content Multi-style Generation and Simplified Pseudo-label Selection"**

**Rating:** 8
**Confidence:** 5

**Review:**

The paper presents a two - stage framework for unsupervised domain adaptation in abdominal MRI segmentation. Its main contribution lies in the single - content multi - style generative network, which captures the intra - domain gap of the target domain and serves as data augmentation. Also, the simplified pseudo - label selection method reduces training complexity. The comments are listed below:
(1) Fig. 1 is too complex to understand. The authors could make it concise for high level ideas or split it to serveral figures.
(2) The loss functions should be explained in details.
(3) There are many typos or formatting issues. In the caption of Figure 2, “My ablation results denotes” should be “My ablation results denote”.
(4) “Our method need train just once instead of three times.” in the "Simplified pseudo-label selection method" should be “Our method needs to train just once instead of three times.”

---

> ### Author Response · Authors · 2025-03-29
> **Response to Reviewer n39Y**
>
> Comment 1: Fig. 1 is too complex to understand. The authors could make it concise for high level ideas or split it to serveral figures.
>
> Thank you for pointing out the limitations of Fig. 1. In response to the reviewer’s comments, we have reorganized the original figure into two distinct figures. The revised Fig. 1 provides a clear overview of the high level ideas, while Fig. 2 specifically details the generative network architecture.
>
> Comment 2: The loss functions should be explained in details.
>
> Thank you for pointing out the need for the details of the loss functions. We have provided additional clarification in Section 2.2.
>
> Comment 3: There are many typos or formatting issues. In the caption of Figure 2, “My ablation results denotes” should be “My ablation results denote”.
>
> Thank you very much for your careful review and for pointing out the typos or formatting issues. Specifically, we have revised the sentence as follows:"Our ablation results denote the small nnU-Net by only training generated CT2MRI scans."
>
> Comment 4: “Our method need train just once instead of three times.” in the "Simplified pseudo-label selection method" should be “Our method needs to train just once instead of three times.”.
>
> Thank you very much for your careful review and for pointing out the typos or formatting issues. We sincerely apologize for this oversight. Upon reviewing the manuscript, we have made the necessary correction as per your suggestion. Specifically, we have revised the sentence as follows:"Our method needs train just once instead of three times."

---

### Decision · Program_Chairs · 2025-03-20

**Decision:**

Accept

**Comment:**

Please carefully address the reviewers' comments in the revision.

---

> ### Author Response · Authors · 2025-03-29
> **Response to Program Chairs**
>
> Thank you for your feedback. We have carefully addressed all the reviewers' comments and revised the manuscript accordingly. A detailed response to each point is included in the revised submission, and the requested test results have been incorporated into the manuscript.